# Television Rating Control in the Multichannel Environment Using Trend Fuzzy Knowledge Bases and Monitoring Results[†]

**Olexiy Azarov [1], Leonid Krupelnitsky [1] and Hanna Rakytyanska [2,*]**

[1] Computer Facilities Department, Vinnytsia National Technical University, 95, Khmelnitske sh., 21021 Vinnytsia, Ukraine; azarov2@vntu.edu.ua (O.A.); krupost@gmail.com (L.K.)

[2] Soft Ware Design Department, Vinnytsia National Technical University, 95, Khmelnitske sh., 21021 Vinnytsia, Ukraine

[*] Correspondence: h_rakit@ukr.net; Tel.: +380-432-61-68-02

[†] This paper is an extended version of Azarov, O.; Krupelnitsky, L.; Rakytyanska, H. A fuzzy model of television rating control with trend rules tuning based on monitoring results. In Proceedings of the 2018 IEEE Second International Conference on Data Stream Mining & Processing (DSMP); Lviv, Ukraine, 21–25 August 2018, pp. 369–375. ISBN: 978-1-5386-8175-6.

**Abstract:** The purpose of this study is to control the ratio of programs of different genres when forming the broadcast grid in order to increase and maintain the rating of a channel. In the multichannel environment, television rating controls consist of selecting content, the ratings of which are completely restored after advertising. The hybrid approach to rule set refinement based on fuzzy relational calculus simplifies the process of expert recommendation systems construction. By analogy with the problem of the inverted pendulum control, the managerial actions aim to retain the balance between the fuzzy demand and supply. The increase or decrease trends of the demand and supply are described by primary fuzzy relations. The rule-based solutions of fuzzy relational equations connect significance measures of the primary fuzzy terms. Program set refinement by solving fuzzy relational equations allows avoiding procedures of content-based selective filtering. The solution set generation corresponds to the granulation of television time, where each solution represents the time slot and the granulated rating of the content. In automated media planning, generation of the weekly TV program in the form of the granular solution provides the decrease of time needed for the programming of the channel broadcast grid.

**Keywords:** TV channel rating; expert recommendation systems; inverted pendulum control; fuzzy classification knowledge bases; solving fuzzy relational equations

## 1. Summary

The top priority task of TV company personnel is to control the ratio of programs of different genres when forming the broadcast grid in order to increase and maintain the rating of the channel [1]. Intelligent recommendation systems in the TV domain are based on the classification rules connecting the time factors and the users' preferences with the TV ratings [2,3]. An automatic recommendation scheme based on collaborative filtering infers the preferred TV programs in two stages [4]. At the first stage, a candidate set of programs is generated on the grounds of the users' watching history [5]. At the second stage, the candidate TV programs are ranked to eliminate the recommendation redundancy stipulated by items similarity [4].

Selection of the preferred programs from the candidate program set is carried out by content-based collaborative filtering. The selective filtering recommends items on the grounds of a comparison

between the items content and the viewers' demand. The aim of program selection is to reduce the system complexity by detecting similar items and merging multiple preferences in the candidate rule set [4,6]. At the collaborative filtering stage, the dimensionality reduction technique, singular value decomposition, is used to find the most similar items in each time and item cluster [7,8]. In context-based filtering, similarity measures are defined using ontology [8]. To reduce computational time, item metadata are clustered according to the item genre [9]. Papers [10,11] present the metrics to measure similarity between groups of users. Finally, these criteria are incorporated into the genetic algorithm of selective collaborative filtering [10,11].

In neural computing, the rule mining problem is solved using the traditional "find all—then prune and select" approach [12,13]. Rule generation using the support vector machine methodology provides enhanced recommendation accuracy [14]. To identify fine-grained time and content preferences, a support vector machine (SVM) separates and classifies the data via hyperboxes. The undesirable effect of hyperboxes overlapping is associated with the traditional problem of rule refinement selection.

The television domain has become increasingly complex due to the multichannel environment. Television advertising time is purchased on the basis of the projected future ratings [15]. The problem lies in the paradoxical connection between the programs ratings before and after the ad unit [16]. In the multichannel environment, the ad skipping behaviors may result in lower ratings [17]. Restoration of the program rating after the ad unit depends on the decision to watch or skip the program; i.e., give preference to the programs broadcast by the concurrent channels. The conventional recommender systems concentrate on the targeted advertising and do not take the ad break-factor into account [18,19]. In this case, the skipping phenomenon stipulated by the multichannel environment is ignored. Such methods of media planning will inevitably reduce the rating points because of the content sensitivity to the advertisement insertion [19].

In this paper, a model of television rating control is built on the grounds of the method of nonlinear dependencies identification by fuzzy knowledge bases [20]. The problem of retaining the television rating is similar to the inverted pendulum control with the aim of retaining it in a vertical position. This problem can be successfully solved using fuzzy logic [20,21]. By analogy with the problem of the inverted pendulum control, the managerial actions aim to retain the balance between the fuzzy demand and supply. In work [22], the fuzzy control model, which provides the balanced demand and supply for each content category is proposed. A control action is formed, which consists of increasing or decreasing the rating of the programs in the channel broadcast grid according to the distribution of viewers' demand for the programs of different genres [22].

Unlike [22], the fuzzy control model is expanded on the account of the supplementary factors influencing the demand and supply. The viewers' demand for different content categories is defined by the time factors; i.e., the time of viewing and the day of the week. In the multichannel environment, a television rating control consists of selecting such content, the ratings of which are completely restored after advertising. Therefore, projected ratings after control in each genre category are evaluated by the content attributes and the ad break-factor. In this case, targeted advertising is augmented by the viewer intention to watch the program. Finally, the content category for each time slot is recommended on the grounds of users' preferences.

Since there is no satisfactory methodical standard for the optimal refinement of the program set, television time is granulated by generating a candidate set of daily TV programs with further content-based selective filtering. At the same time, experts establish the trends of demand-supply relationships, which can be described by primary fuzzy relations. In this case, the increase or decrease of the television indices can be considered as the primary fuzzy terms. The solution to the problem of the program set refinement may be the use of fuzzy relational calculus [23,24]. The rule-based solutions of fuzzy relational equations connect significance measures of the primary fuzzy terms and reflect the semantic intensiveness of the increase or decrease trend [25–27]. Resolution of fuzzy relational equations for the given decision classes guarantees the optimal number of rules or time slots and

the optimal granularity of the content attributes. Recommendation accuracy is achieved through the complete solution set; i.e., the complete rule set of the recommendation knowledge base [28,29].

The composite fuzzy model of the control action balancing the demand and supply was proposed in [22]. In this paper, the composite fuzzy models for the demand and supply are proposed. The composite fuzzy model of the demand defined by the time factors allows generating item and timing recommendations simultaneously. The composite fuzzy model of the supply defined by the content and advertisement attributes provides the rating restoration in the multichannel environment. The hybrid approach to rule set refinement based on fuzzy relational calculus [24] simplifies the process of expert recommendation systems construction. Program set refinement by solving fuzzy relational equations allows avoiding procedures of content-based selective filtering [25–27]. The solution set generation corresponds to the granulation of the television time, where each solution represents the time slot and the granulated rating of the content. In automated media planning, generation of the weekly TV program in the form of the granular solutions provides the decrease of time needed for the programming of the channel broadcast grid.

Following the approach proposed in [25–27], the genetic algorithm is used for tuning the primary fuzzy model and solving the primary system of fuzzy relational equations [28,29]. The genetic algorithm [28,29] transforms the initial relational equations into solvable ones. The formation of the complete solution set is accomplished by the exact analytical methods supported by the free software [24]. Linguistic interpretation, i.e., selection of input terms, lies in the maximum approximation to the partition by interval solutions of the primary equation system [27]. Finally, parametric tuning of the linguistic solutions is accomplished in accordance with work [30].

## 2. Data Description

The fuzzy model was constructed using the example of the television channel Inter, which holds leading positions in the Ukrainian media market [31]. The TV channel Inter presents programs of such basic genres: political programs and news releases ($k = 1$); TV serials and documentary projects ($k = 2$); entertaining and sports programs ($k = 3$).

For the TV rating control problem, the monitoring and forecast window will be for one week. We shall denote the time and day of the week for the TV program release as $\mathbf{t} = (t_1, t_2)$. Management is carried out at the level of each air-hour, i.e., $t_1 \in [0, 24]$, $t_2 \in [1, 7]$. Let p be the number of the current week for media planning. The proportion of TV viewers who watch TV at the time moment ($\mathbf{t}$, p) determines the rating of the TV channel. For the channel Inter, the ratings reach 20%. The analysis of the TV channel rating is carried out according to the results of monitoring of the TV programs ratings obtained for the previous week p − 1. The TV program is compiled for the forthcoming week p.

The timing and item recommendations in the form of the weekly TV program is presented in [32]. The weekly TV program is compiled as a grid with the following nodes: the date (the time of viewing and the day of the week) and the attributes of the program (the genre and title of the program).

The popularity estimates of the TV programs included in the weekly broadcast grid can be obtained from the Internet Movie Data Base (imdb) [33]. This data set contains nearly 5000 titles of the top rated TV shows in 26 genre categories. Imdb-ratings of the TV shows produced in Ukraine are collected in [34]. In developing the fuzzy control model, we shall use the imdb-ratings as the supply attributes weighted using the scale from 0 to 10 [33,34].

Effectiveness of time and content management is evaluated by the weekly top 20 rating [31]. The archive of weekly ratings contains data from 2012–2018 years. The weekly top 20 can be represented as a table with the following columns: the title of the program, the date and time, the rating and share. The weekly top 20 list contains the ratings influenced by the ad break-factor. The ad break-factor can be considered as a proportion of viewers who return to the program after advertising. TV experts connect the ad break-factor with the lowering down of the possible ratings up to 40%.

Ratings of the programs, which do not fall into the weekly top 20 list are evaluated as follows. It is obvious that the ratings of these programs do not exceed the minimum value among the most

popular weekly programs. For the newly launched TV shows, the imdb-ratings can be temporarily substituted by the actual ratings from the weekly top 20 list.

## 3. Method of the Recommendation System Construction

### 3.1. Structure of the TV Rating Control Model

The structure of the TV rating control model corresponds to the following hierarchical tree of logic inference:

- for content management

$$x_k = f_x^k(t_1, t_2, p), \ k = \overline{1, n}, \tag{1}$$

$$y_k(\mathbf{t}, p) = f_y^k(x_k(\mathbf{t}, p), z_k(\mathbf{t}, p - 1)), \tag{2}$$

- for rating evaluation

$$a_k(\mathbf{t}, p) = f_a^k(x_k(\mathbf{t}, p)), \tag{3}$$

$$v_k(\mathbf{t}, p) = f_v^k(z_k(y_k(\mathbf{t}, p)), a_k(\mathbf{t}, p)), \tag{4}$$

$$u(\mathbf{t}, p) = f_u(v_1(\mathbf{t}, p)), \dots, v_n(\mathbf{t}, p)), \tag{5}$$

where

n is the number of genres of the TV programs;

$x_k(\mathbf{t}, p)$ is the viewers' demand for the programs of the genre k at the time moment $(\mathbf{t}, p)$;

$z_k(\mathbf{t}, p - 1)$ is the imdb-rating of the program of the genre k at the time moment $(\mathbf{t}, p - 1)$;

$y_k(\mathbf{t}, p)$ is a control action for the time moment $(\mathbf{t}, p)$, consisting in increasing–decreasing the imdb-rating of the program of the genre k;

$z_k(y_k(\mathbf{t}, p))$ is the imdb-rating of the program offered after the control action;

$a_k(\mathbf{t}, p)$ is the break-factor of the program of the genre k with the advertisement at the time moment $(\mathbf{t}, p)$;

$v_k(\mathbf{t}, p)$ is the rating of the program of the genre k restored after the ad unit at the time moment $(\mathbf{t}, p)$;

$u(\mathbf{t}, p)$ is the rating of the TV channel at the time moment $(\mathbf{t}, p)$.

It is supposed that the control action is determined as the difference between the imdb-rating values before and after control, i.e., $y_k(\mathbf{t}, p) = z_k(\mathbf{t}, p) - z_k(\mathbf{t}, p - 1)$.

Variation ranges of the TV indices are defined as follows: [0, 10] points for $x_k$ and $z_k$; [–10, 10] points for $y_k$; [0.4, 1] for $a_k$; [0, 20] % for $v_k$ and u.

We shall describe the trend dependencies with the help of the primary fuzzy terms:

- in the morning (M), in the afternoon (A), in the evening (Ev) for $t_1$;
- on weekdays (Wd), on weekends (We) for $t_2$;
- increased (decreased) (I, D) or stable (St) for $x_k$, $z_k$, $v_k$ or u;
- increase (decrease) (I, D) or stay inactive (N) for $y_k$.

For the composite knowledge base construction, we shall use the linguistic modifiers: sharply (sh), moderately (m), weakly (w). These modifiers describe the semantic intensity of the primary terms D and I [25–27].

Functional dependencies (1)–(5) are defined by the primary fuzzy relations presented in Tables 1–5.

It is necessary to transfer the primary fuzzy relations into the composite fuzzy rules for the modified decision classes of the variables $x_k(\mathbf{t}, p)$, $y_k(\mathbf{t}, p)$, $a_k(\mathbf{t}, p)$, $v_k(\mathbf{t}, p)$ and $u(\mathbf{t}, p)$. The composite rules were built for the seven classes (sh-m-wD, St, w-m-shI) of the variables $x_k(\mathbf{t}, p)$, $a_k(\mathbf{t}, p)$, $v_k(\mathbf{t}, p)$, $u(\mathbf{t}, p)$; for the seven classes (sh-m-wD, N, w-m-shI) of the variable $y_k(\mathbf{t}, p)$.

**Table 1.** Primary fuzzy relations "viewing time—demand".

| IF | | THEN $x_k(t,p)$ | | |
|---|---|---|---|---|
| | | D | St | I |
| $t_1$ | M | w-m | m | w-m |
| | A | m | m-sh | m |
| | Ev | w | m-sh | sh |
| $t_2$ | Wd | m-sh | sh | m-sh |
| | We | m | m-sh | m-sh |

**Table 2.** Primary fuzzy relations "demand and supply—control action".

| IF | | THEN $y_k(t,p)$ | | |
|---|---|---|---|---|
| | | D | N | I |
| $x_k(t,p)$ | D | sh | sh | w |
| | St | m-sh | m-sh | sh |
| | I | w | m | sh |
| $z_k(t,p-1)$ | D | w | sh | sh |
| | St | m-sh | m-sh | m |
| | I | sh | m | w |

**Table 3.** Primary fuzzy relations "demand—ad break-factor".

| IF | | THEN $a_k(t,p)$ | | |
|---|---|---|---|---|
| | | D | St | I |
| $x_k(t,p)$ | D | m-sh | w-m | w |
| | St | w-m | m-sh | w-m |
| | I | w | m | sh |

**Table 4.** Primary fuzzy relations "supply with the ad break-factor—restored genre ratings".

| IF | | THEN $v_k(t,p)$ | | |
|---|---|---|---|---|
| | | D | St | I |
| $z_k(t,p)$ | D | sh | w | w |
| | St | m | m-sh | m |
| | I | m | m-sh | sh |
| $a_k(t,p)$ | D | sh | w | w |
| | St | m | m-sh | w-m |
| | I | w-m | m-sh | sh |

**Table 5.** Primary fuzzy relations "restored genre ratings—rating of the channel".

| IF | | THEN $u(t,p)$ | | |
|---|---|---|---|---|
| | | D | St | I |
| $v_k(t,p)$ | D | m-sh | w-m | w |
| | St | w-m | m-sh | m-sh |
| | I | w | w-m | sh |

### 3.2. The Problem of Tuning the Fuzzy Control Model

The fuzzy control model connects the vectors of significance measures of the primary fuzzy terms of the variables $t_1$, $t_2$, $x_k$, $z_k$ and $y_k$ in correlations (1) and (2); variables $x_k$, $a_k$, $z_k$, $v_k$ and u in correlations (3)–(5). The vectors of significance measures of the primary fuzzy terms can be considered as the vectors of the fuzzy causes and effects.

Correlations (1)–(5) define the primary fuzzy model in the form:

- for content management

$$\mu_x^k = (\mu_t^1 \circ H_1^k) \cap (\mu_t^2 \circ H_2^k), \tag{6}$$

$$\mu_y^k = (\mu_x^k \circ \mathbf{Q}_1^k) \cap (\mu_z^k \circ \mathbf{Q}_2^k), \tag{7}$$

- for rating evaluation

$$\mu_a^k = \mu_x^k \circ \mathbf{G}^k, \tag{8}$$

$$\mu_v^k = (\mu_Z^k \circ \mathbf{R}_1^k) \cap (\mu_a^k \circ \mathbf{R}_2^k), \tag{9}$$

$$\mu_u = (\mu_v^1 \circ \mathbf{W}^1) \cup \ldots \cup (\mu_v^n \circ \mathbf{W}^n), \tag{10}$$

where

$\mu_t^1 = (\mu_t^{1,1}, \ldots, \mu_t^{1,3})$ and $\mu_t^2 = (\mu_t^{2,1}, \mu_t^{2,2})$ are the vectors of the fuzzy causes M, A, Ev and Wd, We for the variables $t_1$ and $t_2$;

$\mu_x^k = (\mu_x^{k,1}, \ldots, \mu_x^{k,3})$ and $\mu_z^k = (\mu_z^{k,1}, \ldots, \mu_z^{k,3})$ are the vectors of the fuzzy causes D, St, I for the variables $x_k$ and $z_k$;

$\mu_y^k = (\mu_y^{k,1}, \ldots, \mu_y^{k,3})$ is the vector of the fuzzy effects D, N, I for the variable $y_k$;

$\mu_a^k = (\mu_a^{k,1}, \ldots, \mu_a^{k,3})$ and $\mu_Z^k = (\mu_Z^{k,1}, \ldots, \mu_Z^{k,3})$ are the vectors of the fuzzy causes D, St, I for the variables $a_k$ and $z_k(y_k)$;

$\mu_v^k = (\mu_v^{k,1}, \ldots, \mu_v^{k,3})$ and $\mu_u = (\mu_u^1, \ldots, \mu_u^3)$ are the vectors of the fuzzy effects D, St, I for the variables $v_k$ and $u$;

$\mathbf{H}_1^k = [h_1^{k,IJ}]$, $\mathbf{H}_2^k = [h_2^{k,LJ}]$ and $\mathbf{Q}_1^k = [q_1^{k,IJ}]$, $\mathbf{Q}_2^k = [q_2^{k,IJ}]$, $I, J = \overline{1,3}$, $L = \overline{1,2}$, are the primary fuzzy relational matrices "viewing time $(t_1, t_2)$—demand $x_k$" and "demand $x_k$ and supply $z_k$—control action $y_k$";

$\mathbf{G}^k = [g^{k,IJ}]$, $I, J = \overline{1,3}$, are the primary fuzzy relational matrices "demand $x_k$—ad break-factor $a_k$";

$\mathbf{R}_1^k = [r_1^{k,IJ}]$, $\mathbf{R}_2^k = [r_2^{k,IJ}]$ and $\mathbf{W}^k = [w^{k,IJ}]$, $I, J = \overline{1,3}$, are the primary fuzzy relational matrices "supply $z_k(y_k)$ with the ad break-factor $a_k$—restored genre ratings $v_k$—rating of the channel $u$";

$\circ$ and $\circ$, $\cap$ are the operations of the simplified and extended max-min composition corresponding to the correlations $f_a^k$, $f_u$ and $f_x^k$, $f_y^k$, $f_v^k$ [23].

Following the composition laws [23], the primary system of fuzzy relational equations is derived from the relations (6)–(10):

- for content management

$$\mu_x^{k,1} = ((\mu_t^{1,1} \wedge h_1^{1,11}) \vee \ldots \vee (\mu_t^{1,3} \wedge h_1^{1,31})) \wedge ((\mu_t^{2,1} \wedge h_2^{2,11}) \vee (\mu_t^{2,2} \wedge h_2^{2,21}));$$
$$\ldots \tag{11}$$
$$\mu_x^{k,3} = ((\mu_t^{1,1} \wedge h_1^{1,13}) \vee \ldots \vee (\mu_t^{1,3} \wedge h_1^{1,33})) \wedge ((\mu_t^{2,1} \wedge h_2^{2,13}) \vee (\mu_t^{2,2} \wedge h_2^{2,23}));$$

$$\mu_y^{k,1} = ((\mu_x^{k,1} \wedge q_1^{k,11}) \vee \ldots \vee (\mu_x^{k,3} \wedge q_1^{k,31})) \wedge ((\mu_z^{k,1} \wedge q_2^{k,11}) \vee \ldots \vee (\mu_z^{k,3} \wedge q_2^{k,31}));$$
$$\ldots \tag{12}$$
$$\mu_y^{k,3} = ((\mu_x^{k,1} \wedge q_1^{k,13}) \vee \ldots \vee (\mu_x^{k,3} \wedge q_1^{k,33})) \wedge ((\mu_z^{k,1} \wedge q_2^{k,13}) \vee \ldots \vee (\mu_z^{k,3} \wedge q_2^{k,33}));$$

- for rating evaluation

$$\mu_a^{k,1} = (\mu_x^{k,1} \wedge g^{k,11}) \vee \ldots \vee (\mu_x^{k,3} \wedge g^{k,31});$$
$$\ldots \tag{13}$$
$$\mu_a^{k,3} = (\mu_x^{k,1} \wedge g^{k,13}) \vee \ldots \vee (\mu_x^{k,3} \wedge g^{k,33});$$

$$\mu_v^{k,1} = ((\mu_Z^{k,1} \wedge r_1^{k,11}) \vee \ldots \vee (\mu_Z^{k,3} \wedge r_1^{k,31})) \wedge ((\mu_a^{k,1} \wedge r_2^{k,11}) \vee \ldots \vee (\mu_a^{k,3} \wedge r_2^{k,31}));$$
$$\ldots \tag{14}$$
$$\mu_v^{k,3} = ((\mu_Z^{k,1} \wedge r_1^{k,13}) \vee \ldots \vee (\mu_Z^{k,3} \wedge r_1^{k,33})) \wedge ((\mu_a^{k,1} \wedge r_2^{k,13}) \vee \ldots \vee (\mu_a^{k,3} \wedge r_2^{k,33}));$$

$$\mu_u^1 = ((\mu_v^{1,1} \wedge w^{1,11}) \vee \ldots \vee (\mu_v^{1,3} \wedge w^{1,31})) \vee \ldots \vee ((\mu_v^{n,1} \wedge w^{n,11}) \vee \ldots \vee (\mu_v^{n,3} \wedge w^{n,31}));$$
$$\ldots \tag{15}$$
$$\mu_u^3 = ((\mu_v^{1,1} \wedge w^{1,13}) \vee \ldots \vee (\mu_v^{1,3} \wedge w^{1,33})) \vee \ldots \vee ((\mu_v^{n,1} \wedge w^{n,13}) \vee \ldots \vee (\mu_v^{n,3} \wedge w^{n,33})).$$

In "single input—single output" subsystems of the Equations (11)–(15), the operations $\vee$ and $\wedge$ are replaced by max and min [23]. In the case of the "multiple inputs—single output" system, the subsystems can be connected by the operation max for the simplified composition [20] or by the operation min for the extended composition law [23].

The primary fuzzy terms are described using the bell-shaped membership function model of variable $\tau$ to arbitrary term T in the form [20,21]:

$$\mu^T(\tau) = 1/(1 + ((\tau - \beta)/\sigma)^2), \tag{16}$$

where $\beta$ is a coordinate of function maximum, $\mu^T(\beta) = 1$; $\sigma$ is a parameter of concentration.

The model (16) is used for evaluation of the fuzzy causes and effects significance measures.

In this case, correlations (6)–(10) take the form:

- for content management

$$\mu_x^k = f_x^k(t_1, t_2, p, \mathbf{H}_1^k, \mathbf{H}_2^k, \mathbf{\Psi}_t^1, \mathbf{\Psi}_t^2), \tag{17}$$

$$\mu_y^k(y_k, \mathbf{\Psi}_y) = f_y^k(\mu_x^k, z_k, \mathbf{Q}_1^k, \mathbf{Q}_2^k, \mathbf{\Psi}_z^k), \tag{18}$$

- for rating evaluation

$$\mu_a^k = f_a^k(\mu_x^k, \mathbf{G}^k), \tag{19}$$

$$\mu_v^k = f_v^k(\mu_Z^k, \mu_a^k, \mathbf{R}_1^k, \mathbf{R}_2^k, \mathbf{\Psi}_z^k), \tag{20}$$

$$\mu_u(u, \mathbf{\Psi}_u) = f_u(\mu_v^1, \ldots, \mu_v^n, \mathbf{W}^1, \ldots, \mathbf{W}^n), \tag{21}$$

where

$\mathbf{\Psi}_t^1 = (\beta_t^{1,1}, \sigma_t^{1,1}, \ldots, \beta_t^{1,3}, \sigma_t^{1,3})$, $\mathbf{\Psi}_t^2 = (\beta_t^{2,1}, \sigma_t^{2,1}, \beta_t^{2,2}, \sigma_t^{2,2})$ and $\mathbf{\Psi}_z^k = (\beta_z^{k,1}, \sigma_z^{k,1}, \ldots, \beta_z^{k,3}, \sigma_z^{k,3})$ are the vectors of parameters of the primary membership functions of the input variables $t_1$, $t_2$ and $z_k$;

$\mathbf{\Psi}_y^k = (\beta_y^{k,1}, \sigma_y^{k,1}, \ldots, \beta_y^{k,3}, \sigma_y^{k,3})$ and $\mathbf{\Psi}_u = (\beta_u^1, \sigma_u^1, \ldots, \beta_u^3, \sigma_u^3)$ are the vectors of parameters of the primary membership functions of the output variables $y_k$ and $u$.

It is assumed that some training data sample can be obtained on the grounds of successful managerial decisions

$$\left\langle \hat{p}, \; \hat{t}_{kl}^{\wedge P}, \; \hat{z}_{kl}^P(\hat{t}_{kl}^{\wedge P}, \hat{p} - 1), \; \hat{z}_{kl}^P(\hat{t}_{kl}^{\wedge P}, \hat{p}), \; \hat{u}_{kl}^P(\hat{t}_{kl}^{\wedge P}, \hat{p}) \right\rangle, \; p = \overline{1, P}, \; l = \overline{1, L_{pk}},$$

where P is the number of weeks in the data sample;

$L_{pk}$ is the number of TV programs of the genre k in the weekly experiment number p;

$\hat{t}_{kl}^{\wedge P}$ and $\hat{z}_{kl}^P$ are the control system state parameters in the experimental time slot number pkl;

$\hat{u}_{kl}^P$ is the TV rating in the experimental time slot $\hat{t}_{kl}^{\wedge P}$.

The essence of the fuzzy models (17)–(21) tuning is as follows. It is necessary to find the relation matrices $\mathbf{H}_{1-2}^k, \mathbf{Q}_{1-2}^k, \mathbf{G}^k, \mathbf{R}_{1-2}^k, \mathbf{W}^k$ and the vectors of the membership functions parameters $\mathbf{\Psi}_t^1, \mathbf{\Psi}_t^2, \mathbf{\Psi}_z^k, \mathbf{\Psi}_y^k, \mathbf{\Psi}_u$, which provide the minimum distance between theoretical and experimental data:

$$\sum_{p=1}^P \sum_{k=1}^n \sum_{l=1}^{L_{pk}} [f_u(\hat{p}, \hat{t}_{kl}^P, \hat{z}_{kl}^P, \mathbf{H}_{1-2}^k, \mathbf{Q}_{1-2}^k, \mathbf{G}^k, \mathbf{R}_{1-2}^k, \mathbf{W}^k, \mathbf{\Psi}_t^{1-2}, \mathbf{\Psi}_z^k, \mathbf{\Psi}_y^k) - \hat{\mu}_u(\hat{u}_{kl}^P, \mathbf{\Psi}_u)]^2 = \min_{\mathbf{H,Q,G,R,W,\Psi}}. \tag{22}$$

We shall denote:

$X_{kj}$ and $Y_{kj}$, $j = \overline{1, 7}$, are the modified decision classes of the variables $x_k(\mathbf{t}, p)$ and $y_k(\mathbf{t}, p)$;

$A_{kj}$, $V_{kj}$ and $U_j$, $j = \overline{1, 7}$, are the modified decision classes of the variables $a_k(\mathbf{t}, p)$, $v_k(\mathbf{t}, p)$ and $u(\mathbf{t}, p)$, respectively;

$N_x^{kj}$ and $N_y^{kj}$ are the numbers of composite rules in the classes $X_{kj}$ and $Y_{kj}$;

$N_a^{kj}$, $N_v^{kj}$ and $N_u^j$ are the numbers of composite rules in the classes $A_{kj}$, $V_{kj}$ and $U_j$, respectively.

Given the output classes, the solution set of primary fuzzy relational Equations (6)–(10) can be considered as the set of composite fuzzy rules [25–27]:

- for content management

$$\underset{i=\overline{1,N_x^{kj}}}{\cup} \left[ \left[ \underline{\mu}_t^{1,ij}, \overline{\mu}_t^{1,ij} \right] \cap \left[ \underline{\mu}_t^{2,ij}, \overline{\mu}_t^{2,ij} \right] \right] \to x_k = X_{kj}; \tag{23}$$

$$\underset{i=\overline{1,N_y^{kj}}}{\cup} \left[ \left[ \underline{\mu}_x^{k,ij}, \overline{\mu}_x^{k,ij} \right] \cap \left[ \underline{\mu}_z^{k,ij}, \overline{\mu}_z^{k,ij} \right] \right] \to y_k = Y_{kj}; \tag{24}$$

- for rating evaluation

$$\underset{i=\overline{1,N_a^{kj}}}{\cup} \left[ \underline{\mu}_x^{k,ij}, \overline{\mu}_x^{k,ij} \right] \to a_k = A_{kj}; \tag{25}$$

$$\underset{i=\overline{1,N_v^{kj}}}{\cup} \left[ \left[ \underline{\mu}_Z^{k,ij}, \overline{\mu}_Z^{k,ij} \right] \cap \left[ \underline{\mu}_a^{k,ij}, \overline{\mu}_a^{k,ij} \right] \right] \to v_k = V_{kj}; \tag{26}$$

$$\underset{i=\overline{1,N_u^j}}{\cup} \left[ \left[ \underline{\mu}_v^{1,ij}, \overline{\mu}_v^{1,ij} \right] \cup \ldots \cup \left[ \underline{\mu}_v^{n,ij}, \overline{\mu}_v^{n,ij} \right] \right] \to u = U_j. \tag{27}$$

In (23)–(27), the fuzzy solution vectors are presented in the form of the vectors of the lower and upper bounds of the fuzzy causes significance measures, where $\underline{\mu}_t^{1,ij} = (\underline{\mu}_t^{11,ij}, \ldots, \underline{\mu}_t^{13,ij})$, $\overline{\mu}_t^{1,ij} = (\overline{\mu}_t^{11,ij}, \ldots, \overline{\mu}_t^{13,ij})$ and $\underline{\mu}_t^{2,ij} = (\underline{\mu}_t^{21,ij}, \underline{\mu}_t^{22,ij})$, $\overline{\mu}_t^{2,ij} = (\overline{\mu}_t^{21,ij}, \overline{\mu}_t^{22,ij})$ are the lower and upper fuzzy solution vectors for the variables $t_1$ and $t_2$; $\underline{\mu}_x^{k,ij} = (\underline{\mu}_x^{k1,ij}, \ldots, \underline{\mu}_x^{k3,ij})$, $\overline{\mu}_x^{k,ij} = (\overline{\mu}_x^{k1,ij}, \ldots, \overline{\mu}_x^{k3,ij})$ and $\underline{\mu}_z^{k,ij} = (\underline{\mu}_z^{k1,ij}, \ldots, \underline{\mu}_z^{k3,ij})$, $\overline{\mu}_z^{k,ij} = (\overline{\mu}_z^{k1,ij}, \ldots, \overline{\mu}_z^{k3,ij})$ are the lower and upper fuzzy solution vectors for the variables $x_k$ and $z_k$; $\underline{\mu}_Z^{k,ij} = (\underline{\mu}_Z^{k1,ij}, \ldots, \underline{\mu}_Z^{k3,ij})$, $\overline{\mu}_Z^{k,ij} = (\overline{\mu}_Z^{k1,ij}, \ldots, \overline{\mu}_Z^{k3,ij})$ and $\underline{\mu}_a^{k,ij} = (\underline{\mu}_a^{k1,ij}, \ldots, \underline{\mu}_a^{k3,ij})$, $\overline{\mu}_a^{k,ij} = (\overline{\mu}_a^{k1,ij}, \ldots, \overline{\mu}_a^{k3,ij})$ are the lower and upper fuzzy solution vectors for the variables $z_k(y_k)$ and $a_k$; $\underline{\mu}_v^{k,ij} = (\underline{\mu}_v^{k1,ij}, \ldots, \underline{\mu}_v^{k3,ij})$, $\overline{\mu}_v^{k,ij} = (\overline{\mu}_v^{k1,ij}, \ldots, \overline{\mu}_v^{k3,ij})$ are the lower and upper fuzzy solution vectors for the variables $v_k$.

Given the primary fuzzy model (6) and (7) and the output classes $X_{kj}$ and $Y_{kj}$, $j = \overline{1,7}$, the problem of tuning the composite fuzzy model for content management is formulated as follows [25–27]. For each output class and $Y_{kj}$, $j = \overline{1,7}$, the solution set (23), (24) should be found which provides the least distance between observed and model fuzzy effects vectors in correlations (6) and (7):

$$\left[ f_y^k(\mu_x^k, \mu_z^k, \mathbf{Q}_{1-2}^k) - \mu_y^k(Y_{kj}) \right]^2 = \underset{\mu_x^k, \mu_z^k}{\min}, \tag{28}$$

$$\left[ f_x^k(\mu_t^1, \mu_t^2, \mathbf{H}_{1-2}^k) - \mu_x^k(X_{kj}) \right]^2 = \underset{\mu_t^1, \mu_t^2}{\min}. \tag{29}$$

Given the primary fuzzy models (8)–(10) and the output classes $A_{kj}$, $V_{kj}$ and $U_j$, $j = \overline{1,7}$, the problem of tuning the composite fuzzy model for rating evaluation is formulated as follows [25–27]. For each output class $A_{kj}$, $V_{kj}$ and $U_j$, $j = \overline{1,7}$, the solution set (25)–(27) should be found which provides the least distance between observed and model fuzzy effects vectors in correlations (8)–(10):

$$\left[ f_u(\mu_v^1, \ldots, \mu_v^n, \mathbf{W}^1, \ldots, \mathbf{W}^n) - \mu_u(U_j) \right]^2 = \underset{\mu_v^1, \ldots, \mu_v^n}{\min}, , \tag{30}$$

$$\left[ f_v^k(\mu_Z^k, \mu_a^k, \mathbf{R}_{1-2}^k) - \mu_v^k(V_{kj}) \right]^2 = \underset{\mu_Z^k, \mu_a^k}{\min}, \tag{31}$$

$$[f_a^k(\boldsymbol{\mu}_x^k, \mathbf{G}^k) - \mu_a^k(A_{kj})]^2 = \min_{\boldsymbol{\mu}_x^k}. \qquad (32)$$

The genetic algorithm is used for solving the optimization problems (22), (28)–(32) of tuning the primary fuzzy model and rule-based solutions of the primary fuzzy relational equations [28,29].

Exact analytical methods for the lower and upper solutions of fuzzy relational equations are presented in [24]. The genetic algorithm [28] transforms the initial relational Equations (6)–(10) into solvable ones by finding a null solution of the optimization problems (28)–(32). In this case, the transformed fuzzy effects vector corresponds to the exact null solution and provides the analytical solvability of the initial system. Formation of the complete solution set for the transformed fuzzy effects vectors is accomplished by the exact analytical methods supported by the free software [24].

To evaluate the performance of the resulting rule set, the linguistic interpretation of the granular solutions (23)–(27) is required. Since the number of rules is already known, selection of input terms lies in the maximum approximation to the partition by interval solutions of the primary equation system. Linguistic interpretation of the resulting solutions is performed by the relational partition of the space of input variables to retain the level of detail and the density of coverage [27]. Finally, parametric tuning of the linguistic solutions is accomplished in accordance with work [30].

## 4. Results of Tuning the Fuzzy Control Model

### 4.1. Results of the Primary Fuzzy Model Tuning

The fuzzy control model was tuned using the data presented by the TV channel Inter [31]. The training sample includes data from 2015 to 2018. The trend of the ratings lowering down due to the multichannel environment has been observed since 2015.

The training sample includes data for 170 weeks: 27 weeks in 2018; 46 weeks in 2017; 48 weeks in 2016; 49 weeks in 2015. The broadcast grid is programmed for weekdays and weekends. At the level of each air-hour, the daily broadcast grid is divided by 17 rated time slots from 7.00 to 24.00. Thus, the total number of training pairs is $170 \times 2 \times 17 = 5780$. These data are obtained using $170 \times 20 = 3400$ TV programs, the ratings of which are included in the weekly top 20 list [31].

The values $\langle$viewers' demand $\hat{x}_k(\mathbf{t}, p)$, supply before and after control $\hat{z}_k(\mathbf{t}, p-1)$, $\hat{z}_k(\mathbf{t}, p)$, control action $\hat{y}_k(\mathbf{t}, p) = \hat{z}_k(\mathbf{t}, p) - \hat{z}_k(\mathbf{t}, p-1)$, rating $\hat{u}(\mathbf{t}, p)\rangle$, corresponding to the experienced manager actions, were taken as the training data sample. In this case, the TV rating was maintained at a consistently high level, and the unmet viewers' demand was reduced to a minimum. The unmet demand after control in each genre category can be defined as $\Delta_k(\mathbf{t}, p) = -z_k(\mathbf{t}, p)$. It is supposed that for the balanced demand $x_k(\mathbf{t}, p) = z_k(\mathbf{t}, p)$. The experimental values $\hat{z}_k(\mathbf{t}, p-1)$, $\hat{z}_k(\mathbf{t}, p)$ and $\hat{u}(\mathbf{t}, p)$ are determined on the basis of the weekly top 20 rating [31]. The experimental $x_k(\mathbf{t}, p)$ values $\hat{x}_k(\mathbf{t}, p)$ for each genre category are defined as the ratio of viewers who watch the programs of the genre k broadcast by all TV channels at the time moment $(\mathbf{t}, p)$ [22]. Evaluation of the TV programs ratings in the channel broadcast grid is carried out with the help of the authors' monitoring system [35].

In (11), (13), and (15), the trend fuzzy relations are tuned for each content category. In (12) and (14), the trend fuzzy relations are tuned regardless of the genre. The results of the primary fuzzy model tuning are presented in Appendix A. Parameters of the membership functions for the input and output primary fuzzy terms are presented in Tables A1 and A2. The primary fuzzy relational matrices after tuning are presented in Tables A3–A7.

### 4.2. Solving Fuzzy Relational Equations

Let us consider the construction of the composite fuzzy rules (23)–(27).

The results of the composite fuzzy model tuning are presented in Appendix B. In Tables A8–A12, the sets of solutions in the form of the fuzzy causes vectors correspond to the fuzzy effects vectors

obtained for the given decision classes. Bounds of the decision classes $U_j$ and $Y_{kj}$, $j = \overline{1,7}$, were defined as follows:

$$[\underline{u}, \overline{u}] = \underbrace{[0, 3)}_{\text{shD}} \cup \underbrace{[3, 6)}_{\text{mD}} \cup \underbrace{[6, 9)}_{\text{wD}} \cup \underbrace{[9, 11)}_{\text{St}} \cup \underbrace{[11, 13)}_{\text{wI}} \cup \underbrace{[13, 17)}_{\text{mI}} \cup \underbrace{[17, 20]}_{\text{shI}},$$

$$[\underline{y_k}, \overline{y}_k] = \underbrace{[-10, -7)}_{\text{shD}} \cup \underbrace{[-7, -3)}_{\text{mD}} \cup \underbrace{[-3, -1)}_{\text{wD}} \cup \underbrace{[-1, 1)}_{\text{N}} \cup \underbrace{[1, 3)}_{\text{wI}} \cup \underbrace{[3, 7)}_{\text{mI}} \cup \underbrace{[7, 10]}_{\text{shI}}.$$

For the given bounds of decision classes, the fuzzy effects vectors $\mu_u(U_j)$ and $\mu_y^k(Y_{kj})$ were defined with the help of the primary membership functions of the variables u and $y_k$ (Table A2).

For the decision classes $V_{kj}$, $A_{kj}$ and $X_{kj}$, $j = \overline{1,7}$, the fuzzy effects vectors $\mu_v^k(V_{kj})$, $\mu_a^k(A_{kj})$ and $\mu_x^k(X_{kj})$ were defined by the fuzzy solution vectors obtained for the higher levels of the hierarchical tree of logic inference.

The sets of interval rules corresponding to the sets of solutions from Tables A8–A12 are presented in Tables 6–10. Linguistic interpretation of the obtained solutions allows generating the hierarchical composite fuzzy rules. For the simplified and extended composition laws [23], the sets of solutions are interpreted in the form of the "single input—single output" (Tables 6 and 8) and "multiple inputs—single output" (Tables 7, 9 and 10) rules [28,29]. The lower and upper bounds of the interval rules were obtained with the help of the primary membership functions of the variables $t_1$, $t_2$ and $z_k$ (Table A1).

**Table 6.** Composite fuzzy rules for the TV rating.

| IF (or) | | | THEN |
|---|---|---|---|
| $v_1(t,p)$ | $v_2(t,p)$ | $v_3(t,p)$ | $u(t,p)$ |
| 0–2.0, shD | 0–4.0, shD | 0–9.0, shD–wD | shD |
| 2.0–4.4, shD–mD | 3.8–4.4, mD | 0–9.0, shD–wD | mD |
| 4.4–7.4, mD–wD | 4.4–7.4, mD–wD | 6.2–11.2, wD–St | wD |
| 6.8–9.0, wD | 7.6–10.0, wD–St | 6.2–11.2, wD–St | St |
| 8.4–12.9, St–wI | 10.2–12.9, wI | 8.8–14.0, St–wI | wI |
| 10.8–16.7, wI–mI | 12.4–15.6, wI–mI | 10.8–15.6, wI–mI | mI |
| 16.7–20, shI | 15.6–20, mI–shI | 15.6–20, mI–shI | shI |

**Table 7.** Composite fuzzy rules for the restored genre ratings.

| IF (and) | | THEN |
|---|---|---|
| $z_k(y_k(t,p))$ | $a_k(t,p)$ | $v_k(t,p)$ |
| 0–1.2 | shD | shD |
| 1.2–2.2 | shD | mD |
| 0–2.2 | mD | |
| 2.2–3.7 | mD–wD | wD |
| 3.5–4.4 | wD–St | |
| 2.9–4.4 | St–wI | St |
| 3.7–5.0 | wD–wI | |
| 5.0–6.9 | wD–St | |
| 5.1–6.9 | wI–mI | wI |
| 5.9–7.7 | St–wI | |
| 6.9–7.9 | wI–mI | mI |
| 5.4–7.6 | mI–shI | |
| 7.9–10 | shI | shI |

**Table 8.** Composite fuzzy rules for the ad break-factor.

|  | **IF (or)** |  | **THEN** |
| --- | --- | --- | --- |
| $x_1(t,p)$ | $x_2(t,p)$ | $x_3(t,p)$ | $a_k(t,p)$ |
| shD–mD | shD | shD | shD |
| mD | mD | mD | mD |
| mD–wD | mD–wD | mD–wD | wD |
| wD–St | wD–wI | St–wI | St |
| wI–mI | wI | wI–mI | wI |
| St–mI | mI | wI–mI | mI |
| shI | shI | shI | shI |

**Table 9.** Composite fuzzy rules for the control action.

|  | **IF (and)** | **THEN** |
| --- | --- | --- |
| $x_k(t,p)$ | $z_k(t,p-1)$ | $y_k(t,p)$ |
| shD | 8.3–10 | shD |
| mD | 7.6–10 | |
| mD–shD | 6.1–7.6 | |
| mD–shD | 3.6–5.9 | mD |
| wD–wI | 7.6–10 | |
| mD–shD | 1.8–2.6 | |
| wD–wI | 5.9–6.2 | |
| wD | 3.4–6.2 | wD |
| mI | 7.1–10 | |
| shD | 0–1.8 | |
| St | 3.8–5.0 | N |
| shI | 7.9–10 | |
| mD | 0–2.1 | |
| wD–wI | 3.4–3.5 | |
| wI | 3.50–5.8 | wI |
| mI–shI | 7.4–7.9 | |
| wD–wI | 0–1.9 | |
| mI–shI | 3.4–5.3 | |
| mI | 0–1.9 | mI |
| mI–shI | 1.9–2.8 | |
| shI | 0–1.4 | shI |

**Table 10.** Composite fuzzy rules for the viewers' demand.

|  | **IF (and)** |  |  |  |  | **THEN** |
| --- | --- | --- | --- | --- | --- | --- |
| **k = 1** | | **k = 2** | | **k = 3** | | |
| $t_1$ | $t_2$ | $t_1$ | $t_2$ | $t_1$ | $t_2$ | $x_k(t,p)$ |
| 12–14 | 1–5 | 5–8 | 1–5 | 11–14 | 1–5 | shD |
| 10–12 | 1–5 | 5–8 | 6, 7 | 10–12 | 1–5 | mD |
| 14–16 | 1–5 | 8–11 | 1–5 | | | |
| 11–17 | 6,7 | | | | | |
| 5–7 | 1–5 | 8–10 | 6, 7 | 8–10 | 1–7 | wD |
| 18–20 | 6, 7 | 10–12 | 1–5 | | | |
| 22–24 | 1–5 | | | | | |
| 7–8 | 1–7 | | | 5–8 | 1–7 | |
| 19–21 | 6, 7 | 12–16 | 1–7 | 12–16 | 6, 7 | St |
| 21–22 | 1–7 | | | 14–16 | 1–5 | |
| 8–11 | 6, 7 | 16–18 | 1–7 | 9–12 | 6, 7 | wI |
| 16–18 | 1–7 | | | 16–18 | 1–7 | |
| 8–10 | 1–5 | 17–19 | 6, 7 | 17–20 | 1–7 | mI |
| 18–20 | 1–5 | 18–24 | 1–5 | 18–24 | 1–5 | |
| 20–21 | 1–5 | 18–24 | 6, 7 | 19–24 | 6, 7 | shI |

### 4.3. Example of the TV Rating Control: Construction of the Weekly TV Program

The training sample fragment is presented in Tables 11 and 12 in the form of the weekly TV program constructed for weekdays and weekends from 1.10.2018 to 7.10.2018. Media planning is accompanied with the analysis of the main television indices. Tables 11 and 12 reflect the dynamics of the demand and supply change for each genre during the day. The experimental demand $\hat{x}_k(t,p)$ and supply $\hat{z}_k(t,p-1)$ are balanced by the model control action $y_k(t,p)$. To balance the demand, the priority content category is chosen for each time slot. The priority genre k is in high demand compared to other categories. The obtained pairs "time slot **t**—imdb-rating $z_k(t,p)$ after control" represent the broadcast grid of the forthcoming week. The characteristics of the generated content are evaluated by the ad break-factor $a_k(t,p)$. Stability of the content to the ad break-factor guarantees the high television ratings $v_k(t,p)$. Planning without alternatives is based on the past behavior of viewers. In this case, monitoring results depict the rating restored after the ad unit; i.e., $u(t,p)=v_k(t,p)$. In the case of the alternative propositions $v_k(t,p)$, the preliminary selection is based on user preferences. Finally, the model $u(t,p)$ and experimental $\hat{u}(t,p)$ ratings are compared for each time slot.

It is shown from Tables 11 and 12 that the weekly top 20 list covers each time slot. For the political genre, the offered programs have balanced the viewers' demand. On weekdays from 18 to 23 h, some popular serials have been offered instead of the entertainment programs. On weekends, the demand for the genre of television serials from 18 to 23 h has been satisfied with the programs of the sports and entertainment genre.

A comparison of the model and experimental ratings for $p \in [1, 32]$ weeks is shown in Figure 1. Figure 1 depicts the dynamics of the average weekly rating change at the level of each air-hour on weekdays and weekends from autumn 2017 to spring 2018. When compiling the average weekly rating, the time range is $t_1 \in [8, 23]$ hours, since the programs in the range $t_1 \in [0, 7]$ do not fall into the weekly top 20 rating.

**Table 11.** Weekdays timetable.

| Time Slot | Television Indices for Genre k | | | | | | | |
|---|---|---|---|---|---|---|---|---|
| | $\hat{x}_k(t,p)$ | $\Delta_k(t,p)$ | $\hat{z}_k(t,p-1)$ | $y_k(t,p)$ | $z_k(t,p)$ | $a_k(t,p)$ | $v_k(t,p)/u(t,p)$ | $\hat{u}(t,p)$ |
| 7–8 | 1, 3, St | 2, shD | | | | | | |
| 8–9 | 1, mI; | 2, mD | 1, 3, 5.6 | 0 | 1, 3, 5.6 | 1, 3, 0.99 | 1, 3, 14.0 | 1, 3, 15.0 |
| 9–10 | 3, wD | | | | | | | |
| 10–11 | 2, wD | 1, 3, mD | 2, 4.5 | 0 | 2, 4.5 | 2, 0.84 | 2, 5.5 | 2, 4.6 |
| 11–12 | | | | | | | | |
| 12–13 | 1, 2, St | 3, shD | 1, 3.9 | 0 | 1, 3.9 | 1, 0.89 | 1, 4.3 | 1, 4.5 |
| 13–14 | | | | 0 | 2, 4.8 | 2, 0.85 | 2, 9.0 | 2, 7.8 |
| 14–15 | 2, 3, St | 1, mD | 2, 4.8 | 2, +0.7 | 2, 5.5 | 2, 0.88 | 2, 11.0 | 2, 9.5 |
| 15–16 | | | | | | | | |
| 16–17 | 2, 3 wI | 1, wI | 2, 5.9 | 2, +0.5 | 2, 6.4 | 2, 0.89 | 2, 12.9 | 2, 12.1 |
| 17–18 | | | 1, 6.2 | 0 | 1, 6.2 | 1, 0.85 | 1, 12.5 | 1, 13.0 |
| 18–19 | 2, 3, mI | 1, mI | 2, 8.6 | 0 | 2, 8.6 | 2, 0.99 | 2, 16.5 | 2, 14.0 |
| 19–20 | | | | | | | | |
| 20–21 | 1, shI | 2, 3, mI | 1, 8.3 | 0 | 1, 8.3 | 1, 0.98 | 1, 12.0 | 1, 10.5 |
| 21–22 | 2, 3, mI | 1, St | | 2, +1.3 | 2, 8.0 | 2, 0.87 | 2, 11.5 | 2, 7.5 |
| 22–23 | 2, 3, mI | 1, wD | 2, 6.7 | | | | | 2, 6.8 |
| 23–24 | | | 2, 6.1 | 2, +1.3 or 3, +8.1 | 2, 7.4 or 3, 8.1 | 2, 0.86 or 3, 0.72 | 2, 9.0 or 3, 8.5 | |

**Table 12.** Weekends timetable.

| Time Slot | Television Indices for Genre k | | | | | | | |
|---|---|---|---|---|---|---|---|---|
| | $\hat{x}_k(t,p)$ | $\Delta_k(t,p)$ | $\hat{z}_k(t,p{-}1)$ | $y_k(t,p)$ | $z_k(t,p)$ | $a_k(t,p)$ | $v_k(t,p)/u(t,p)$ | $\hat{u}(t,p)$ |
| 7–8 | 1, 3, St | 2, mD | 1, 3, 4.4 | 0 | 1, 3, 4.4 | 1, 3, 0.85 | 1, 3, 5.5 | 1, 3, 4.5 |
| 8–9 | 1, wI; 3, wD | 2, wD | | | | | | |
| 9–10 | | | 1, 3, 3.1 | 1, 3, +1.9 | 1, 3, 5.0 | 1, 3, 0.90 | 1, 3, 10.0 | 1, 3, 8.1 |
| 10–11 | 1, 3, wI | 2, wD | 3, 6.5 | 2, +4.8 or 3, 0 | 2, 4.8 or 3, 6.5 | 2, 0.81 or 3, 0.84 | 2, 7.2 or 3, 8.5 | 3, 6.1 |
| 11–12 | 2, wD; 3, wI | 1, mD | | | | | | |
| 12–13 | 2, 3, St | 1, mD | 2, 5.8 | 2, −0.3 | 2, 5.5 | 2, 0.79 | 2, 11.0 | 2, 9.0 |
| 13–14 | | | | | | | | |
| 14–15 | | | | 2, +0.4 | 2, 6.2 | 2, 0.81 | 2, 11.5 | 2, 8.6 |
| 15–16 | | | | | | | | |
| 16–17 | 2, 3, wI | 1, mD | 2, 6.9 | 2, +0.5 | 2, 7.4 | 2, 0.75 | 2, 8.2 | 2, 5.5 |
| 17–18 | 2, wI | 1, 3, wI | | | | | | |
| 18–19 | 2, 3 mI | 1, wD | 3, 8.8 | 2, +9.1 or 3, 0 | 2, 9.1 or 3, 8.8 | 2, 0.82 or 3, 0.90 | 2, 12.5 or 3, 12.9 | 3, 8.2 |
| 19–20 | 2, shI; 3, mI | 1, wD | | | | | | |
| 20–21 | 1, St; 2, 3, shI | 2, 3, shI | 1, 6.7 | 0 | 1, 6.7 | 1, 0.90 | 1, 8.3 | 1, 7.6 |
| 21–22 | | 1, St | 3, 9.1 | 2, +9.3 or 3, 0 | 2, 9.3 or 3, 9.1 | 2, 0.95 or 3, 0.99 | 2, 12.0 or 3, 12.2 | 3, 8.6 |
| 22–23 | 2, 3, shI | 1, wD | 3, 8.7 | | | | | |
| 23–24 | | | | 0 | 3, 8.7 | 3, 0.79 | 3, 9.0 | 3, 5.6 |

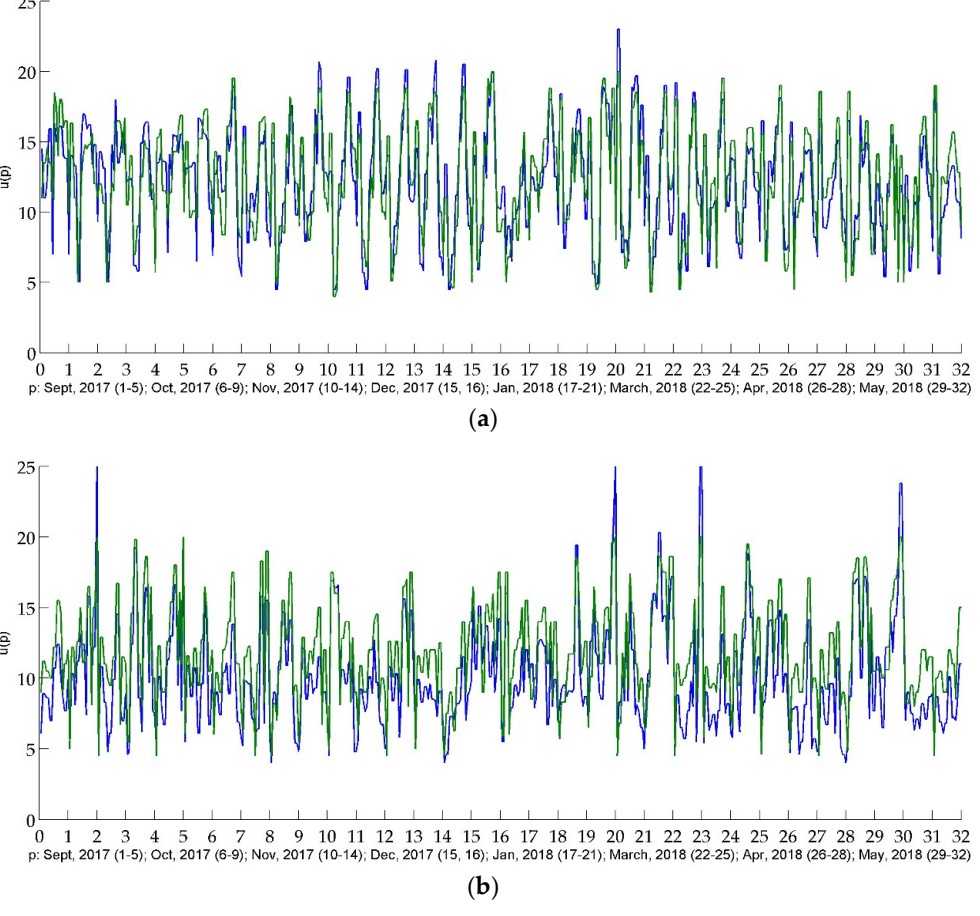

**Figure 1.** The dynamics of the average weekly TV rating change for the model and experimental control action: (**a**) on weekdays; (**b**) on weekends. The model is green, the experimental is blue.

The tuning time for this method is 45 min. The fuzzy control model ensures the correctness of the rating evaluation at the level of RMSE = 1.8847 for weekdays; RMSE = 2.5428 for weekends. On weekdays, the TV ratings are less sensitive to the multichannel environment. The ratings are sufficiently high and the evaluation error decreases. On weekends, the multichannel environment influences the TV ratings significantly. The ratings are unstable and the evaluation error increases.

The limitation of this method is the format of the control action. It is presumed that the viewers' demand can be satisfied immediately. In managerial decision making, the control action heavily depends on the purchase prices of new programs and the expected advertising revenue.

## 5. Effectiveness Estimation of the Hybrid Approach

The weekly TV program set was refined using the methods of genetic tuning applied to the linguistic models enhanced with SVD [3,7,20]; min-max neural networks [12]; SVM [14]. The given method simplifies the process of the program set refinement by solving the primary system of fuzzy relational equations. The effectiveness estimates of the hybrid approach proposed are given below.

We shall denote:

n is the number of input parameters (genres);
T and M are the numbers of input and output primary fuzzy terms;
Z is the number of composite fuzzy rules.

Candidate rule generation corresponds to the construction of a zero option of the timetable. In works [3,7,12,14,20], candidate rule generation requires solving the optimization problem with $2nZ$ variables for boundaries of interval rules. Application of the hybrid approach allows reducing the number of tuning parameters by solving $Z$ optimization problems for $2T$ boundaries of significance measures [25–27]. Rule generator tuning is the optimization problem with $2T+2M+TM$ variables for the trend relational matrix and membership functions parameters.

Selection, that is, finding the best configuration of zero option terms and rules, corresponds to the granulation of the television time. As a result, the broadcast grid can be represented in the form of the time slots and the granulated ratings of the content.

In content-based filtering, selection requires solving the optimization problem with $nZ$ variables. A term selection sign with the possibility of merging the similar terms, as well as the degree of the rule relevance, are subject to tuning [3,7,12,14,20]. In the case of $Z$ solutions of the trend system of equations, the selection is reduced to maintaining the level of detail and the density of coverage. Application of the hybrid approach reduces the number of tuning parameters by solving $Z$ optimization problems for the $T$ modified terms in each rule [25–27].

The problem of TV rating control is reduced to the parallel programming of the television time for six genres (news and analytics, TV serials, documentary projects, programs for children, sports, entertainment). The permissible value of the planning time-frame is 1 h. The results of tuning for different methods are given in Table 13.

**Table 13.** Comparison of the methods for the program set refinement.

| Method of Program Set Refinement | RMSE | | Tuning Time, min |
|:---:|:---:|:---:|:---:|
| | Weekdays | Weekends | |
| Linguistic models + SVD | 1.9425 | 2.6148 | 79 |
| Min-max neural networks | 1.9086 | 2.5509 | 68 |
| SVM | 1.8915 | 2.5376 | 72 |
| Solving fuzzy relational equations | 1.8847 | 2.5428 | 45 |

It is shown from Table 13 that the time of tuning the fuzzy control model according to the methods [3,7,12,14,20] exceeds the planning time-frame (Intel Core i5 LGA1151 3.5 GHz processor).

The tuning time for the given method allows media planning at the level of each air-hour without a loss of recommendation accuracy.

## 6. Conclusions

The fuzzy model for retaining the TV rating is proposed in the framework of fuzzy relational calculus. By analogy with the problem of inverted pendulum control, the managerial actions aim to retain the balance between the fuzzy demand and supply in the multichannel environment. The demand-supply trends are described by the primary fuzzy relations, where the increase or decrease of the television indices is described by the primary fuzzy terms. The refined fuzzy model is built using the linguistic modifiers of the primary fuzzy terms. The time factors, content and advertisement attributes are taken into account to balance the demand and supply. The fuzzy model of the demand allows generating item and timing recommendations simultaneously. The fuzzy model of the supply provides restoration of the ratings in the multichannel environment.

The method of expert recommendation systems construction for TV domain is proposed. TV experts define the primary fuzzy relational matrices. Automated media planning is carried out by solving the primary system of fuzzy relational equations. The granulated television time is determined by the rule-based solution set. The program attributes for each time slot are determined by significance measures of the primary fuzzy terms. The recommendation accuracy is achieved through the complete rule-based solution set. Weekly refinement of the recommended program set by solving fuzzy relational equations allows avoiding procedures of content-based selective filtering. Generation of the weekly TV program in the form of the granular solutions provides the decrease of the computational costs needed for the programming of the television time.

The proposed approach can find application in the automated recommendation systems operating off-line. Further development of this approach can be done in the direction of creating adaptive control models, which are tuned to predict the popularity of the items recommended for on-line TV. In this case, the stage of expert rules refinement should be augmented by a hybrid genetic and neural approach to tuning the primary fuzzy model and solving the primary system of fuzzy relational equations. Besides that, supplementary factors influencing the control actions (purchase prices of new programs, advertising revenue) can be taken into account with the help of the primary fuzzy relations.

**Author Contributions:** Conceptualization, A.O. and K.L.; methodology, A.O., K.L. and R.H.; software, K.L. and R.H.; validation, A.O., K.L. and R.H.; formal analysis, R.H.; investigation, K.L. and R.H.; resources, K.L.; data curation, K.L.; writing—original draft preparation, R.H.; writing—review and editing, R.H.; visualization, K.L. and R.H.; supervision, A.O.; project administration, A.O.; funding acquisition, A.O. and K.L.

**Funding:** This research received no external funding.

**Acknowledgments:** The paper was prepared within the 58–D–369 "Technologies of the construction of intelligent analog-digital systems for monitoring and analysis of multimedia information" project.

**Conflicts of Interest:** The authors declare no conflict of interest. The funders had no role in the design of the study; in the collection, analyses, or interpretation of data; in the writing of the manuscript, or in the decision to publish the results.

## Appendix A

Results of the Primary Fuzzy Model Tuning

**Table A1.** Parameters of the membership functions for the input primary fuzzy terms.

| Parameter | $t_1$ | | | $t_2$ | | $z_k$ | | |
|---|---|---|---|---|---|---|---|---|
| | **M** | **A** | **Ev** | **Wd** | **We** | **D** | **St** | **I** |
| $\beta$ | 5 | 14 | 22 | 1 | 6 | 0.51 | 4.43 | 9.12 |
| $\sigma$ | 3.92 | 3.19 | 3.75 | 4.52 | 0.94 | 2.67 | 1.28 | 2.54 |

**Table A2.** Parameters of the membership functions for the output primary fuzzy terms.

| Parameter | $y_k$ | | | u | | |
|---|---|---|---|---|---|---|
| | D | N | I | D | St | I |
| β | −9.53 | 0.41 | 9.60 | 2.14 | 9.22 | 16.70 |
| σ | 5.15 | 2.76 | 4.89 | 3.70 | 2.56 | 4.39 |

**Table A3.** Primary fuzzy relations "viewing time—demand" after tuning.

| IF | | THEN $x_1(t,p)$ | | | THEN $x_2(t,p)$ | | | THEN $x_3(t,p)$ | | |
|---|---|---|---|---|---|---|---|---|---|---|
| | | D | St | I | D | St | I | D | St | I |
| $t_1$ | M | 0.11 | 0.75 | 0.67 | 0.64 | 0.50 | 0.16 | 0.35 | 0.65 | 0.48 |
| | A | 0.79 | 0.62 | 0.54 | 0.22 | 0.75 | 0.51 | 0.59 | 0.82 | 0.67 |
| | Ev | 0.10 | 0.51 | 0.80 | 0.14 | 0.83 | 1.0 | 0.18 | 0.77 | 0.94 |
| $t_2$ | Wd | 0.80 | 0.91 | 1.0 | 0.61 | 0.85 | 0.68 | 0.64 | 0.89 | 0.67 |
| | We | 0.56 | 0.70 | 0.59 | 0.27 | 0.92 | 1.0 | 0.19 | 0.86 | 1.0 |

**Table A4.** Primary fuzzy relations "demand and supply—control action" after tuning.

| IF | | THEN $y_k(t,p)$ | | |
|---|---|---|---|---|
| | | D | N | I |
| $x_k(t,p)$ | D | 0.98 | 0.81 | 0.12 |
| | St | 0.75 | 0.68 | 0.95 |
| | I | 0.07 | 0.62 | 0.90 |
| $z_k(t, p-1)$ | D | 0.14 | 0.85 | 0.97 |
| | St | 0.83 | 0.70 | 0.65 |
| | I | 0.91 | 0.63 | 0.18 |

**Table A5.** Primary fuzzy relations "demand—ad break-factor" after tuning.

| IF | | THEN $a_k(t,p)$ | | |
|---|---|---|---|---|
| | | D | St | I |
| $x_1(t,p)$ | D | 0.65 | 0.52 | 0.41 |
| | St | 0.26 | 0.82 | 0.76 |
| | I | 0.19 | 0.75 | 0.93 |
| $x_2(t,p)$ | D | 0.88 | 0.24 | 0.10 |
| | St | 0.43 | 0.59 | 0.25 |
| | I | 0.16 | 0.71 | 0.90 |
| $x_3(t,p)$ | D | 0.90 | 0.32 | 0.09 |
| | St | 0.27 | 0.81 | 0.67 |
| | I | 0.11 | 0.60 | 0.99 |

**Table A6.** Primary fuzzy relations "supply with the ad break-factor—restored genre ratings" after tuning.

| IF | | THEN $v_k(t,p)$ | | |
|---|---|---|---|---|
| | | D | St | I |
| $z_k(y_k(t,p))$ | D | 1.0 | 0.12 | 0 |
| | St | 0.56 | 0.80 | 0.58 |
| | I | 0.42 | 0.78 | 0.95 |
| $a_k(t,p)$ | D | 0.93 | 0.08 | 0 |
| | St | 0.50 | 0.67 | 0.33 |
| | I | 0.39 | 0.70 | 0.81 |

**Table A7.** Primary fuzzy relations "restored genre ratings—rating of the channel" after tuning.

| IF | | THEN u(t,p) | | |
|---|---|---|---|---|
| | | D | St | I |
| $v_1(\mathbf{t}, p)$ | D | 0.96 | 0.20 | 0 |
| | St | 0.45 | 0.84 | 0.57 |
| | I | 0 | 0.63 | 0.90 |
| $v_2(\mathbf{t}, p)$ | D | 0.75 | 0.26 | 0 |
| | St | 0.34 | 0.87 | 0.54 |
| | I | 0 | 0.59 | 0.83 |
| $v_3(\mathbf{t}, p)$ | D | 0.69 | 0.30 | 0 |
| | St | 0.33 | 0.76 | 0.51 |
| | I | 0 | 0.50 | 0.82 |

## Appendix B

Results of the Composite Fuzzy Model Tuning

**Table A8.** Solutions of fuzzy relational equations for the TV rating.

| Genre | IF $\mu_v^k$ | | | THEN $\mu_u(U_j)$ |
|---|---|---|---|---|
| | D | St | I | |
| k = 1<br>k = 2<br>k = 3 | 0.96–1<br>0.75–1<br>0.30–1 | 0–0.21 | 0–0.21 | shD,<br>(0.97, 0.40, 0.21) |
| k = 1<br>k = 2<br>k = 3 | 0.72<br>0.25–0.72<br>0.30–1 | 0.25 | 0.25 | mD,<br>(0.72, 0.56, 0.25) |
| k = 1, 2<br>k = 3 | 0–0.45 | 0.30–0.80<br>0.51–1 | 0.27 | wD,<br>(0.43, 0.80, 0.27) |
| k = 1<br>k = 2<br>k = 3 | 0.45 | 0.84–1<br>0.30–0.84<br>0.51–1 | 0–0.51 | St,<br>(0.30, 1, 0.39) |
| k = 1<br>k = 2<br>k = 3 | 0.34 | 0.76<br>0.30–0.76<br>0.76–1 | 0–0.51<br>0–0.51<br>0.57 | wI,<br>(0.21, 0.76, 0.54) |
| k = 1<br>k = 2, 3 | 0.34 | 0–0.63 | 0.74<br>0.34–0.74 | mI,<br>(0.17, 0.50, 0.74) |
| k = 1<br>k = 2, 3 | 0–0.32 | 0–0.32 | 0.90–1<br>0.82–1 | shI,<br>(0.14, 0.32, 0.91) |

**Table A9.** Solutions of fuzzy relational equations for the restored genre ratings.

| | IF | | | | | THEN |
|---|---|---|---|---|---|---|
| | $\mu_Z^k$ | | | $\mu_a^k$ | | $\mu_v^k(V_{kj})$ |
| D | St | I | D | St | I | |
| 0.93–1 | 0.21 | 0–0.21 | 0.93–1 | 0.21 | 0–0.21 | shD, (0.96, 0.21, 0.21) |
| 0.72 0.72–1 | 0.25 0–0.25 | 0–0.25 0.25 | 0.72–1 0.72 | 0.25 0–0.25 | 0–0.25 0.25 | mD, (0.72, 0.25, 0.25) |
| 0.45–1 | 0.80–1 0–0.80 | 0.27–1 0.78–1 | 0–0.50 | 0.67–1 | 0–0.33 | wD, (0.45, 0.80, 0.27) |
| 0.56 0–0.56 0.45 | 0.80–1 0.80–1 0–0.80 | 0–0.58 0.58 0.78–1 | 0–0.50 0–0.50 0–1 | 0.67–1 0.50–1 0.67–1 | 0.58–1 0.70–1 0.51 | St, (0.45, 1, 0.51) |
| 0–1 0–0.42 | 0.76 0–0.42 | 0–0.58 0.76 | 0–0.39 0–1 | 0–0.39 0.67–1 | 0.70–1 0.57 | wI, (0.34, 0.76, 0.57) |
| 0–0.42 0–1 | 0–0.42 0–0.63 | 0.74–1 0.74 | 0–1 0–0.39 | 0.67 0–0.39 | 0.74 0.74–1 | mI, (0.34, 0.63, 0.74) |
| 0–0.32 | 0–0.32 | 0.81–1 | 0–0.39 | 0–0.39 | 0.81–1 | shI, (0.32, 0.32, 0.82) |

**Table A10.** Solutions of fuzzy relational equations for the ad break-factor.

| Genre | IF | | | THEN |
|---|---|---|---|---|
| | $\mu_x^k$ | | | $\mu_a^k(A_{kj})$ |
| | D | St | I | |
| k = 1 k = 2 k = 3 | 0.65–1 0.88–1 0.90–1 | 0–0.21 0–0.24 0.21 | 0–0.21 0–0.24 0–0.21 | shD, (0.93, 0.21, 0.21) |
| k = 1 k = 2, 3 | 0.65–0.72 0.72 | 0.25 | 0–0.25 | mD, (0.72, 0.25, 0.25) |
| k = 1 k = 2 k = 3 | 0.50–0.67 0.50 0.50 | 0–0.41 0.59–0.67 0.67 | 0–0.33 0.33 0–0.33 | wD, (0.50, 0.67, 0.33) |
| k = 1 k = 2 k = 3 | 0.50 | 0.82–1 0.67–1 0.81–1 | 0–0.41 0.51 0–0.51 | St, (0.50, 1, 0.51) |
| k = 1 k = 2 k = 3 | 0.39 0–0.43 0.39 | 0.57–0.67 0.59–0.67 0.67 | 0–0.57 0.57 0–0.67 | wI, (0.39, 0.67, 0.57) |
| k = 1 k = 2 k = 3 | 0.39 0–0.43 0.39 | 0–0.74 0.59 0–0.60 | 0.74 | mI, (0.39, 0.59, 0.74) |
| k = 1, 2 k = 3 | 0–0.39 0–0.27 | 0.39 | 0.81–1 | shI, (0.39, 0.39, 0.81) |

**Table A11.** Solutions of fuzzy relational equations for the control action.

| | IF | | | | | THEN |
|---|---|---|---|---|---|---|
| | $\mu_x^k$ | | | $\mu_z^k$ | | $\mu_y^k(Y_{kj})$ |
| **D** | **St** | **I** | **D** | **St** | **I** | |
| 0.91–1 | 0–0.12 | 0–0.12 | 0–0.14 | 0–0.14 | 0.91–1 | shD, (0.96, 0.17, 0.11) |
| 0.73–1 | 0.16 | 0–0.16 | 0.38 | 0–0.38 | 0.73–1 | mD, (0.73, 0.38, 0.16) |
| 0.73–1 | 0.38 | 0.16 | 0.16 | 0.73–1 | 0–0.38 | |
| 0–0.38 | 0.73–1 | 0.38 | 0–0.16 | 0.16 | 0.73–1 | |
| 0.62–1 | 0.20 | 0–0.20 | 0.62–1 | 0.43 | 0–0.43 | wD, (0.43, 0.62, 0.20) |
| 0.43 | 0.62–1 | 0–0.20 | 0.20 | 0.62–1 | 0–0.43 | |
| 0–0.43 | 0.43 | 0.62–1 | 0–0.20 | 0.20 | 0.62–1 | |
| 0.81–1 | 0.27 | 0–0.27 | 0.81–1 | 0.30 | 0–0.30 | N, (0.30, 0.96, 0.27) |
| 0–0.30 | 0.81–1 | 0–0.27 | 0–0.27 | 0.81–1 | 0–0.30 | |
| 0–0.30 | 0.30 | 0.81–1 | 0–0.27 | 0.27 | 0.81–1 | |
| 0.74–1 | 0.45 | 0–0.45 | 0.74–1 | 0.21 | 0–0.21 | wI, (0.21, 0.74, 0.45) |
| 0–0.21 | 0.68–1 | 0.45 | 0–0.45 | 0.68–1 | 0.21 | |
| 0–0.21 | 0.21 | 0.68–1 | 0–0.45 | 0.45 | 0.68–1 | |
| 0.16 | 0.77–1 | 0–0.40 | 0.77–1 | 0.40 | 0–0.16 | mI, (0.16, 0.40, 0.77) |
| 0–0.16 | 0.40 | 0.65–1 | 0–0.40 | 0.65–1 | 0.16 | |
| 0–0.16 | 0.16 | 0.77–1 | 0.77–1 | 0–0.40 | 0.40 | |
| 0–0.14 | 0–0.21 | 0.90–1 | 0.90–1 | 0–0.21 | 0.14 | shI, (0.11, 0.21, 0.94) |

**Table A12.** Solutions of fuzzy relational equations for the viewers' demand.

| | IF | | | | | THEN |
|---|---|---|---|---|---|---|
| **Genre** | $\mu_t^1$ | | | $\mu_t^2$ | | $\mu_x^k(X_{kj})$ |
| | **M** | **A** | **Ev** | **Wd** | **We** | |
| k = 1 | 0–0.15 | 0.79–1 | 0–0.15 | 0.56–1 | 0–1 | shD, (0.91, 0.16, 0.12) |
| k = 2 | 0.61–1 | 0–0.17 | 0–0.17 | 0.61–1 | 0–1 | |
| k = 3 | 0–0.15 | 0.59–1 | 0–0.15 | 0.81–1 | 0–0.59 | |
| k = 1 | 0–0.38 | 0.62–1 | 0–0.29 | 0.62–1 | 0.56–1 | mD, (0.62, 0.38, 0.20) |
| k = 2 | 0.62 | 0–0.50 | 0–0.20 | 0.61–1 | 0.61–1 | |
| k = 3 | 0–0.38 | 0.62 | 0–0.20 | 0.62–1 | 0–1 | |
| k = 1 | 0.80–1 | 0–0.43 | 0.80–1 | 0.73–1 | 0–0.73 | wD, (0.43, 0.73, 0.38) |
| k = 2 | 0.43 | 0.73–1 | 0–0.14 | 0–0.73 | 0.73–1 | |
| k = 3 | 0.65 | 0–0.41 | 0–0.41 | | | |
| k = 1 | 0.75–1 | 0–0.30 | 0.81–1 | 0.81–1 | 0–0.70 | St, (0.30, 0.81, 0.27) |
| | | | | 0–0.70 | 0.70–1 | |
| k = 2 | 0.30 | 0.75–1 | 0–0.27 | 0.81–1 | 0–0.81 | |
| | | | | 0–0.81 | 0.81 | |
| k = 3 | 0.65–1 | 0.81–1 | 0–0.27 | | 0–0.81 | |
| | | | | | 0.81–1 | |
| k = 1 | 0.68 | 0–0.54 | 0–0.51 | 0.68–1 | 0–0.68 | wI, (0.16, 0.68, 0.45) |
| k = 2 | 0–0.22 | 0.68 | 0–0.51 | 0–0.68 | 0.68–1 | |
| k = 3 | 0.45 | 0.68 | 0–0.45 | | | |
| k = 1 | 0–0.65 | 0–0.40 | 0–0.65 | 0.65–1 | 0–1 | mI, (0.21, 0.40, 0.65) |
| k = 2 | 0–0.22 | 0–0.65 | 0.65–1 | 0.65 | 0–0.65 | |
| k = 3 | 0–0.21 | 0–0.52 | 0.65–1 | 0–0.65 | 0.65–1 | |
| k = 1 | 0–0.18 | 0.18 | 0.80–0.90 | 0.80–1 | 0–0.59 | shI, (0.14, 0.21, 0.80) |
| k = 2 | 0–0.22 | 0–0.51 | 0.80–1 | 0–0.80 | 0.80–1 | |
| k = 3 | 0–0.17 | 0.21 | 0.72–1 | 0–1 | 0.72–1 | |

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
