# Peer review of "Television Rating Control in the Multichannel Environment Using Trend Fuzzy Knowledge Bases and Monitoring Results"

_data, 2018_

Round 1

Reviewer 1 Report

In this paper, a hybrid approach combining the benefits of semantic training and fuzzy relational equations in simplification of the expert recommendation systems construction is proposed.

In general, authors present a hybrid approach as the combination of semantic training and fuzzy relational equations. Authors should consider the following comments to clarify the main contributions of their paper.      

1.- In the page 2, in the introduction, authors say "The problem of retaining the television rating can be attributed to the problems of fuzzy resources control [1, 19].", in this part, they should include references [a], [b], [c], [d], [e], [f] which also consider the fuzzy resources control.

[a] Neural network updating via argument Kalman filter for modeling of Takagi-Sugeno fuzzy models, Journal of Intelligent & Fuzzy Systems, Vol. 35, No. 2, pp. 2585-2596, 2018.

[b] An inequality approach for evaluating decision making units with a fuzzy output, Journal of Intelligent & Fuzzy Systems, Vol. 34, No. 1, pp. 459-465, 2018.

[c] SOFMLS: online self-organizing fuzzy modified least-squares network, IEEE Transactions on Fuzzy Systems, Vol. 17, No. 6, pp. 1296-1309, 2009.

[d] Improved Stabilization Conditions for Nonlinear Systems with Input and State Delays via T-S Fuzzy Model, Mathematical Problems in Engineering, Vol. 2018, pp. 1-14, 2018.

[e] Design of stabilizers and observers for a class of multivariable TS fuzzy models on the basis on new interpolation functions, IEEE Transactions on Fuzzy Systems, Vol. 26, No. 5, pp. 2649-2662, 2018.

[f] Multiple-attribute decision-making method using similarity measures of single-valued neutrosophic hesitant fuzzy sets based on least common multiple cardinality, Journal of Intelligent & Fuzzy Systems, Vol. 34, No. 6, pp. 4203-4211, 2018.

2.- In the page 7, in the equations (6)-(10), authors should clarify if the vectors of fuzzy causes are the membership functions.

3.- In the page 7, in the equations (6)-(10), authors should clarify if the vectors of fuzzy causes are described by triangular, gaussian or other kind of functions.

4.- In the page 7, in the equations (6)-(10), authors should describe more the operations of simplified and extended max-min composition.

5.- In the page 8, authors describe a bell-shaped membership function by one equation, they should include an equations number to this equation.

6.- In the page 14, in the example, if it is possible, authors should compare their method with other previous.

7.- In the page 18, in the conclusions, authors should include some future research.

Author Response

The authors would like to acknowledge the valuable comments of the referees, which contributed to the significant improvement of the paper.

Corrected text is highlighted with green color.

List of corrections:

1. Page 2, line 76-84: to retain the rating of the TV channel, i.e. retain the balance between the fuzzy demand and supply, we resort to help of analogy with the problem of inverted pendulum control. The concept of fuzzy recourses control is deleted from the paper.  

Reference to inverted pendulum control problem:

Page 1, line 36: in key words

Page 1, line 21, 22: in abstract

Page 19, line 483-485: in conclusion section.

2. Page 7, line 218-219: the vectors of fuzzy causes (effects) are the vectors of significance measures of the primary fuzzy terms.

3. Page 8, 9, line 260-264: the model of membership function for the primary fuzzy terms is described.

4. Page 8, line 247-259: the system of fuzzy relational equations is derived using the compositional rule of inference. The operations of simplified and extended composition are described regarding to the SISO and MISO objects. The formulae (13) and (15) with max-min composition are corrected.

5. Page 9, line 264: the equation (16) is added to evaluate fuzzy causes significance measures.

6. Page 18, line 449-452: the methods of program set refinement are selected for comparison.

Page 19: Table 13 is added, which contains the results of comparison.

Page 19, line 478-481: the results of comparison are discussed.

7. Page 20, line 501-508: some future research is added.

Reviewer 2 Report

It is not very clear the software used to compute the results. Figure 1 must be improved, there is difficult to clearly see where is the error between real and forecasted results.

Author Response

The authors would like to acknowledge the valuable comments of the referees, which contributed to the significant improvement of the paper.

Corrected text is highlighted with green color.

List of corrections:

1. Page 12, line 333-338: The application of the free soft for exact analytical methods of solving fuzzy relational equations is described.

Page 3, line 120-122: the same for the introduction section. 

2. Page 18: The color version of Figure  1 is presented.

Reviewer 3 Report

The purpose of the study is to control the ratio of programs of different genres when forming the broadcast grid in order to increase and maintain the rating of the channel. This study is interesting, but there are some issues need to more explain such as, 1. Literature review, especial the weakness and opportunity of current study. 2. Why use the hybrid approach combining the semantic training and fuzzy relational equations? 3. Why use the Control Model? 4. How many data used in this study? 5. How do you approve your approach is well and excellence? 6. Please provide the real data and applied to you approach. 7. Is there any limited in the study? 8. The reference should be updated.

Author Response

The authors would like to acknowledge the valuable comments of the referees, which contributed to the significant improvement of the paper.

Corrected text is highlighted with green color.

List of corrections:

1. Page 2, line 43-65: the literature review is rewritten. The application of the methods of classification rules tuning to the program set refinement is considered.  

Page 3, line 93-95: the weakness of the known methods is highlighted.

Page 3, line 101-102, 110-113: the opportunity of the proposed method.

2. Page 3, line 110-111: the essence of the hybrid approach is defined as a classification rule set refinement using fuzzy relational calculus. The concept of semantic training is deleted from the paper.  

3. Page 2, line 76-84: to retain the rating of the TV channel, i.e. retain the balance between the fuzzy demand and supply, we resort to help of analogy with the problem of inverted pendulum control.

4. Page 12, line 350-354: the size of training data set is given.

5. Page 12, line 339-344: the stages of transformation of the interval solution set to the linguistic rule set are described.

Page 3, line 122-124: the same for the introduction. 

6. Page 12, line 347-349: the chapter 4 is added to describe the primary tuning using the real data presented by the TV channel at the web page.  

7. Page 17, line 439-441: the limitation of the model is given.

8. The references 2-15 are updated regarding the methods of program set refinement.

Round 2

Reviewer 3 Report

The revised manuscript has addressed my concern.